# Isolation and Structure Determination of New Pyrones from *Dictyostelium* spp. Cellular Slime Molds Coincubated with *Pseudomonas* spp.

**DOI:** 10.3390/molecules29092143

**Published:** 2024-05-05

**Authors:** Takehiro Nishimura, Takuya Murotani, Hitomi Sasaki, Yoshinori Uekusa, Hiromi Eguchi, Hirotaka Ishigaki, Katsunori Takahashi, Yuzuru Kubohara, Haruhisa Kikuchi

**Affiliations:** 1Faculty of Pharmacy, Keio University, 1-5-30 Shibakoen, Minato-ku, Tokyo 105-8512, Japan; uekusa-ys@pha.keio.ac.jp; 2Graduate School of Pharmaceutical Sciences, Tohoku University, 6-3, Aza-Aoba, Aramaki, Aoba-ku, Sendai 980-8578, Japan; k.235268@gmail.com (T.M.); sasaki.hitomi@toaeiyo.co.jp (H.S.); hiromi.e@me.com (H.E.); 3Department of Medical Technology, Faculty of Health Science, Gunma Paz University, Takasaki 370-0006, Japan; ishigaki@paz.ac.jp (H.I.); k-takahashi@paz.ac.jp (K.T.); 4Graduate School of Health and Sports Science, Juntendo University, 1-1 Hiraga-gakuendai, Inzai, Chiba 270-1695, Japan; ykuboha@juntendo.ac.jp

**Keywords:** cellular smile molds, natural products, *Pseudomonas* spp., coincubation, α-pyrone, phenolic compound

## Abstract

Cellular slime molds are excellent model organisms in the field of cell and developmental biology because of their simple developmental patterns. During our studies on the identification of bioactive molecules from secondary metabolites of cellular slime molds toward the development of novel pharmaceuticals, we revealed the structural diversity of secondary metabolites. Cellular slime molds grow by feeding on bacteria, such as *Klebsiella aerogenes* and *Escherichia coli,* without using medium components. Although changing the feeding bacteria is expected to affect dramatically the secondary metabolite production, the effect of the feeding bacteria on the production of secondary metabolites is not known. Herein, we report the isolation and structure elucidation of clavapyrone (**1**) from *Dictyostelium clavatum*, intermedipyrone (**2**) from *D. magnum*, and magnumiol (**3**) from *D. intermedium*. These compounds are not obtained from usual cultural conditions with *Klebsiella aerogenes* but obtained from coincubated conditions with *Pseudomonas* spp. The results demonstrate the diversity of the secondary metabolites of cellular slime molds and suggest that widening the range of feeding bacteria for cellular slime molds would increase their application potential in drug discovery.

## 1. Introduction

Cellular slime molds are soil microorganisms that belong to the eukaryotic kingdom Amoebozoa, which is taxonomically distinct from fungi [1,2]. As one of the most famous cellular slime molds owing to its simple developmental patterns and ease of handling, *Dictyostelium discoideum* has been used as a model organism for studying cell and developmental biology. In addition, considering cellular slime molds as a source of natural compounds, we isolated natural compounds with unique structures and biological activities [3,4,5,6,7,8,9]. For example, brefelamide inhibits osteopontin expression [5], and ppc-1 is a promising candidate for antiobesity drugs [9]. In 2005, the genome *D. discoideum* was found to contain approximately 45 polyketide synthase (PKS) gene clusters [1], and another cellular slime mold species, *D. purpureum*, was predicted to have 50 PKS genes [10]. The number of genes in these organisms is higher than that observed in *Streptomyces avermitilis*, which is known to contain abundant secondary metabolites. Thus, we focused on cellular slime molds as a source of natural compounds to identify bioactive compounds in these organisms for the development of new drugs, revealing the structural diversity of their secondary metabolites. However, despite their usefulness, cellular slime molds have been underutilized in drug discovery compared with fungi, actinomycetes, and proteobacteria.

The production of secondary metabolites is highly dependent on the fermentation conditions, which provide the nutrients necessary for growth, such as carbon, nitrogen, and phosphate. Numerous metals such as zinc, iron, and manganese are also essential for bacterial growth, some of which also affect antibiotic production [11]. Cellular slime molds grow by feeding on bacteria, such as *Klebsiella aerogenes* and *Escherichia coli*, without requiring medium components, whereas little is known about their growth by feeding on other bacteria [12,13]. In fact, the effect of feeding bacteria on the production of secondary metabolites is unknown. Recently, our group isolated monochasiol F and G from *D. monochasioides* fed with *K. aerogenes* [3]. These resorcinols contain unusual alkyl chains with a cyclopropane moiety. The origin of this substructure is considered to be lactobacillic acid, which is widely present in some bacteria. This result shows that coincubating bacteria directly affects the second metabolites of cellular slime molds [3]. Here, we describe the structural elucidation of clavapyrone (**1**), intermedipyrone (**2**), and magnumiol (**3**), which were newly isolated from *D. magnum*, *D. clavatum*, and *D. intermedium*, respectively, and cultured with bacteria of the genus *Pseudomonas* instead of the one that is normally used (Figure 1).

## 2. Results

### 2.1. Isolation and Structure Elucidation of Clavapyrone (***1***)

Multicellular fruiting bodies (62.6 g dry weight) of *D. clavatum* TNS-C-220 were cultured on agar plates and coincubated with *P. fluorescens*. After extracting twice with methanol at room temperature, filtration, and evaporation, the resulting extract (15.5 g) was partitioned between ethyl acetate and water. The fraction soluble in ethyl acetate (2.94 g) was separated using a series of column chromatography to afford clavapyrone (**1**; 8.4 mg). High-resolution electron impact mass spectrometry (HR-EIMS; *m/z* 314.1868 [M]^+^) indicated that the molecular formula of **1** was C_20_H_26_O_3_. Table 1 summarizes the ^1^H and ^13^C nuclear magnetic resonance (NMR) data deduced from the heteronuclear multiple quantum correlation (HMQC) spectrum, which revealed the presence of four methyl groups [*δ*_C_ 18.7 (C-5′), *δ*_C_ 25.7 (C-8″), *δ*_C_ 17.8 (C-9″), and *δ*_C_ 16.8 (C-10″)], three methylene groups, including one oxygenated methylene [*δ*_C_ 65.8 (C-1″), *δ*_C_ 39.5 (C-4″), and *δ*_C_ 26.3 (C-5″)], sp^2^ methine groups [*δ*_C_ 89.1 (C-3), *δ*_C_ 100.8 (C-5), *δ*_C_ 119.8 (C-1′), *δ*_C_ 136.5 (C-2′), *δ*_C_ 130.5 (C-3′), *δ*_C_ 136.9 (C-4′), *δ*_C_ 117.2 (C-2″), and *δ*_C_ 123.6 (C-6″)], and four quaternary carbons. ^1^H–^1^H correlation spectroscopy (COSY) revealed the connectivity from C-1′ to C-5′ through C-2′, C-3′, and C-4′, from C-4″ to C-6″ through C-5″, and the spin coupling between the methylene protons H-1″ and H-2″. Meanwhile, correlations from H-3 to C-2, C-4, and C-5, from H-5 to C-3 and C-4, from H-1′ to C-5 and C-6, and from H-2′ to C-6 were observed in the heteronuclear multiple bond correlation (HMBC) spectrum, which confirmed the presence of a 6-alkyl-α-pyrone moiety. The geometries of the double bonds at C-1′and C-3′ were established as 1′*E* and 3′*E*, respectively, on the basis of the couplings between H-1′ and H-2′ (*J*_H-1′, H-2′_ = 15.3 Hz) and between H-3′ and H-4′ (*J*_H-3′, H-4′_ = 15.2 Hz). The remaining structure was determined to be a geraniol moiety by combining the COSY correlations and ^1^H−^13^C long-range couplings from H-8″ and H-9″ to C-6″ and C-7″ and from H-10″ to C-2″, C-3″, and C-4″. The *E*-geometry was assigned for the C-2″ olefin of **1** by comparing the allylic methyl carbon [*δ*_C_ 16.7 (C-10″)] with that of geraniol or nerol [14]. According to the HMBC correlation from H-1″ to C-4, these two partial structures are connected, furnishing 4-*O*-gerany-6-((1′*E*,3′*E*)-penta-1′,3′-dien-1′-yl)-2-pyrone (**1**), as shown in Figure 2.

*D. discoideum*, DiPKS1, also referred to as Steely1, catalyzes the formation of 2-alkyl-pyrone [15,16]. This type I FAS–type III PKS fusion enzyme is common among most cellular slime molds and is presumed to be involved in the synthesis of differentiation-inducing factors [17,18]. This kind of enzyme constructs the α-pyrone skeleton, which then undergoes *O*-geranylation to afford clavapyrone. Although 4-methoxy-6-alkylpyrones have been reported [19], clavapyrone is the first example of a naturally synthesized 4-*O*-geranylated α-pyranoid.

### 2.2. Isolation and Structure Elucidation of Intermedipyrone (***2***)

Multicellular fruiting bodies (45.2 g dry weight) of *D. intermedium* S90506 were cultured on agar plates, coincubated with *P. fluorescens*, and then extracted three times with methanol at room temperature to yield an extract (6.37 g), which was partitioned between ethyl acetate and water. The fraction soluble in ethyl acetate (2.14 g) was separated using a series of column chromatography to afford intermedipyrone (**2**) (1.2 mg). HR-EIMS (*m/z* 332.2358 [M]^+^) indicated that **2** possessed the molecular formula of C_21_H_32_O_3_. The ^1^H and ^13^C NMR data obtained from the HMQC spectrum are shown in Table 2, which indicated the presence of one methyl group [*δ*_C_ 14.1 (C-12′)], an oxygenated methine [*δ*_C_ 79.8 (C-3)], three sp^2^ methine groups [*δ*_C_ 117.9 (C-5), *δ*_C_ 136.1 (C-6), and *δ*_C_ 116.2 (C-7)], a carbonyl carbon [*δ*_C_ 170.0 (C-1)], and twelve methylene groups [*δ*_C_ 32.9 (C-4), *δ*_C_ 34.8 (C-1′), *δ*_C_ 24.8 (C-2′), *δ*_C_ 31.9 (C-10′), *δ*_C_ 22.7 (C-11′), and *δ*_C_ 29.3–29.6 (C-3′–C-9′)]. A hydrogen-bonded proton was observed at 11.0 ppm in the ^1^H NMR spectrum. The ^1^H–^1^H COSY correlations revealed the connectivities of C-5–C-6–C-7 and C-4–C-3–C-1′–C-2′. HMBC correlations from HO-8 to C-7, C-8, and C-8a and from H-5 to C-3a, C-4, and C-8a together with the MS spectrum indicated the presence of a 3-(2-hydroxytetradecyl)phenol moiety. The remaining molecular formula, CO, was used to construct a dihydroisocoumarin skeleton, resulting in the structure of **2** as shown in Figure 3.

The determination of the absolute configuration of **2** was accomplished by comparing the electronic circular dichroism (ECD) spectra and specific optical rotations of asymmetrically synthesized 3-alkyl-3,4-dihydroxy-8-hydroxyisocoumarins, which indicated that the carbon number of 3-alkyl chains in dihydroisocoumarins does not affect the sign of rotation [20,21,22]. The ECD spectrum of **2** showed a negative Cotton effect at 258 nm, and the C-3 position of **2** was determined to have an *R* configuration (Figure 1 and Appendix A). This determination was confirmed by comparing the experimental and calculated ECD spectrum (Appendix A).

### 2.3. Isolation and Structure Elucidation of Magnumiol (***3***)

Multicellular fruiting bodies (41.9 g dry weight) of *D. magnum* C-113 were cultured on agar plates and coincubated with *P. chororaphis*. Methanol extraction three times at room temperature yielded an extract (5.55 g), which was then partitioned between ethyl acetate and water. The fraction soluble in ethyl acetate (1.58 g) was separated using a series of column chromatography, octadecyl silica gel column chromatography, and preparative thin-layer chromatography (TLC) to afford magnumiol (**3**; 0.6 mg). The molecular formula of **3** was assigned as C_23_H_34_O_3_ via high-resolution fast atom bombardment mass spectrometry (HR-FABMS; *m/z* 381.2380 [M]^+^). The ^1^H and ^13^C NMR data obtained from the HMQC spectrum are shown in Table 3, from which the presence of the following groups was deduced: three methyl groups [*δ*_C_ 24.2 (C-8), *δ*_C_ 17.9 (C-14′), and *δ*_C_ 14.4 (C-15′)], six sp^2^ methine groups [*δ*_C_ 115.6 (C-3), *δ*_C_ 134.1 (C-4), *δ*_C_ 122.9 (C-5), *δ*_C_ 131.8 (C-3′), *δ*_C_ 131.7 (C-12′), and *δ*_C_ 124.6 (C-13′)], a carbonyl carbon [*δ*_C_ 162.9 (C-2)], and nine methylene groups, including one oxygenated methylene [*δ*_C_ 71.9 (C-1′), *δ*_C_ 27.8 (C-4′), *δ*_C_ 32.6 (C-11′), and *δ*_C_ 29.2–29.6 (C-5′–C-10′)]. A hydrogen-bonded proton was observed at 11.4 ppm in the ^1^H NMR spectrum. The connectivity of C-3–C-4–C-5 observed in the ^1^H–^1^H COSY spectrum suggests the presence of a 1,2,3-trisubstituted benzene ring. HMBC correlations from HO-2 to C-1, C-2, and C-3, from H_3_-8 to C-1, C-5, and C-6, and from H-3 to C-8 indicated the presence of a 6-methylsalicylate moiety. The alcohol structure condensed with salicylic acid was determined to be 2-methyltetradeca-2,12-dien-1-ol on the basis of the ^1^H–^1^H COSY, HMBC, and MS spectra (Figure 4). The geometries of the double bonds at C-2′ and C-12′ positions were determined from the differences in the chemical shifts due to the γ-effects of shielded allylic carbons in the ^13^C NMR data [23,24]. The signals of allylic carbons of *E*-isomers generally resonate at a lower field than those of the *Z*-isomers in linear compounds [24]. Thus, the double bonds at C-2′ and C-12′ were determined to possess 2′*E* and 12′*E* geometries, respectively, due to the high-field and low-field chemical shift values of the allylic carbons [*δ*_C_ 32.6 (C-11′), *δ*_C_ 17.9 (C-14′), and *δ*_C_ 14.4 (C-15′)]. In nature, 6-methylsalicylic acid is widely produced by fungi [25,26], and salicylate **3** is characterized by a long-chain alkyl alcohol, indicating the diversity of PKSs in cellular slime molds.

### 2.4. Biological Activities of Compounds ***1*** and ***2***

Next, we investigated the biological activities of the isolated compounds **1** and **2**. Compound **1** exhibited moderate antiproliferative activity against human leukemia K562 cells (IC_50_ 17 μM, Figure 5), whereas compound **2** did not inhibit K562 cells, human cervical cancer HeLa cells, and mouse 3T3-L1 fibroblast cells (a model nontransformed cell line) (IC_50_ > 20 μM). Compounds **1** and **2** at concentrations of up to 100 μM did not show apparent antibacterial activities for Gram-positive (*Staphylococcus aureus*) and Gram-negative (*E. coli*) bacteria. Studies on the biological activities of **3** are currently underway.

## 3. Discussion

In this study, we isolated three novel compounds from the fruiting bodies of cellular smile molds cultured with *Pseudomonas* spp. and determined their structures. 6-Alkylpyrone moiety in clavapyrone and 6-methylsalicylic acid in intermedipyrone are predicted to be biosynthesized by type-I iterative polyketide synthases (iPKS), which are typical of fungi [25,26]. Several cellular slime molds of the genus *Dictyostelium* are known to possess abundant polyketide synthase gene clusters [1,15], which are likely activated via coincubation with *Pseudomonas* spp. to biosynthesize the compounds in the present study. Compound **1** possesses a geranylated pyrone structure, which has not been reported to date, and compound **3** exhibits a salicylate with a characteristic branched alkyl chain. Thus, this study demonstrates that cellular slime molds are a promising source of natural products and presents coculture with bacteria as a new method for obtaining novel compounds. Furthermore, since bacteria other than *Pseudomonas* spp. can be used to culture cellular slime molds [12,13], the number of species of cellular slime molds and feeding bacteria will be increased to demonstrate the usefulness of this organism for drug discovery. In addition, studies are currently underway using techniques, such as Molecular Networking [27], to objectively demonstrate differences in extracts due to changes in feeding bacteria, which will accelerate the exploration of novel compounds.

## 4. Materials and Methods

### 4.1. General Methods

Analytical TLC was performed on silica gel 60 F254 (Merck). Silica and octadecyl silica gel column chromatography was conducted using Biotage Sfär Silica High Capacity Duo (Biotage, Uppsala, Sweden) and Biotage Sfär C18 Duo eluted by Isolera (Biotage, Uppsala, Sweden). NMR spectra were recorded on a JEOL ECA-600 spectrometer and a Bruker AVANCE 600 spectrometer. ^1^H and ^13^C NMR chemical shifts are given in parts per million (δ) relative to tetramethylsilane (δ_H_ 0.00) or residual solvent signals (δ_H_ 7.26, δ_C_ 77.0) as internal standards. Mass spectra were measured using JEOL JMS-700 and JMS-DX303 spectrometers. ECD spectra were measured on a J-1100DS spectrometer. Optical rotations were measured using a JASCO P-1030 polarimeter.

### 4.2. Organisms and Culture Conditions

*Dictyostelium clavatum* TNS-C-220 was provided by NBRP Nenkin (https://nenkin.nbrp.jp/ (accessed on 10 April 2024)). Its spores were cultured at 22 °C with *Pseudomonas fluorescens* on A-medium agar plates containing 0.5% glucose, 0.5% polypeptone, 0.05% yeast extract, 0.225% KH_2_PO_4_, 0.137% Na_2_HPO_4_·12H_2_O, 0.05% MgSO_4_·7H_2_O, and 1.5% agar.

*Dictyostelium intermedium* S90506 was provided by NBRP Nenkin (https://nenkin.nbrp.jp/ (accessed on 10 April 2024)). Its spores were cultured at 22 °C with *Pseudomonas fluorescens* on modified MGY-1 medium agar plates containing 0.75% mannitol, 0.15% L-glutamic acid sodium salt, 0.075% yeast extract, 0.5% proteose peptone, 0.225% KH_2_PO_4_, 0.05% Na_2_HPO_4_·12H_2_O, 0.02% MgSO_4_·7H_2_O, and 1.5% agar. The fruiting bodies formed after four days were harvested for extraction.

*Dictyostelium magnum* C-113 was provided by NBRP Nenkin (https://nenkin.nbrp.jp/ (accessed on 10 April 2024)). Its spores were cultured at 22 °C with *Pseudomonas chororaphis* on modified MGY-2 medium agar plates containing 0.9% mannitol, 0.18% L-glutamic acid sodium salt, 0.09% yeast extract, 0.2% proteose peptone, 0.225% KH_2_PO_4_, 0.05% Na_2_HPO_4_·12H_2_O, 0.02% MgSO_4_·7H_2_O, and 1.5% agar. After four days, the obtained fruiting bodies were harvested for extraction.

### 4.3. Isolation of Clavapyrone (***1***)

The fruiting bodies (dry weight 62.6 g) of *D. clavatum* TNS-C-220 were collected after coincubation with *P. fluorescens* on an A-medium agar plate. They were extracted twice with methanol at room temperature to give an extract (15.5 g), which was then partitioned between ethyl acetate and water to yield an ethyl acetate–soluble fraction (2.94 g). The ethyl acetate–soluble fraction was chromatographed over silica gel by eluting with hexane–ethyl acetate mixtures with increasing polarity to afford fraction A using hexane–ethyl acetate (2:1) as the eluent. Fraction A was separated using an octadecyl silica gel column with a water–acetonitrile solvent system, and fraction B was obtained by eluting with water–acetonitrile (1:9). Fraction B was subjected to recycle preparative HPLC using a GPC-T-2000 column (φ 20 mm × 600 mm, YMC Co., Ltd., Kyoto, Japan) and ethyl acetate as the solvent to give clavapyrone (**1**; 8.4 mg). Data for **1** are as follows: colorless amorphous solid; ^1^H NMR and ^13^C NMR spectroscopic data are shown in Table 1; and HR-EIMS *m/z* 314.1868 [M]^+^ (314.1881 calculated for C_20_H_26_O_3_).

### 4.4. Isolation of Intermedipyrone (***2***)

The fruiting bodies (dry weight 45.2 g) of *D. intermedium* S90506 were collected after coincubation with *P. fluorescens* on modified MGY-1 medium agar plates and then extracted three times with methanol at room temperature to give an extract (6.37 g). The extract was partitioned between ethyl acetate and water to yield an ethyl acetate–soluble fraction (2.14 g), which was chromatographed over silica gel using hexane–ethyl acetate mixtures with increasing polarity as the eluent to afford fraction C when eluting with hexane–ethyl acetate (19:1). Fraction C was separated using a silica gel column with a hexane–chloroform solvent system to give fraction D by eluting with hexane–chloroform (1:1). Fraction D was subjected to recycle preparative HPLC using a GPC-T-2000 column (φ 20 mm × 600 mm, YMC Co., Ltd.) and ethyl acetate as the solvent, affording intermedipyrone (**2**; 1.2 mg). Data for **2** are as follows: colorless amorphous solid; [α]24D −20.6 (c 0.15, chloroform); ^1^H NMR and ^13^C NMR spectroscopic data are shown in Table 2; and HR-EIMS *m/z* 332.2358 [M]^+^ (332.2351 calculated for C_21_H_32_O_3_).

### 4.5. Isolation of Magnumiol (***3***)

The fruiting bodies (dry weight 41.9 g) of *D. magnum* C-113 were collected after coincubation with *P. chororaphis* on modified MGY-2 medium agar plates. Methanol extraction three times at room temperature gave an extract (5.55 g), which was partitioned between ethyl acetate and water to yield an ethyl acetate–soluble fraction (1.58 g). The ethyl acetate–soluble fraction was chromatographed over silica gel by eluting with hexane–ethyl acetate mixtures with increasing polarity to afford fraction E when eluting with hexane–ethyl acetate (2:1). Fraction E was subjected to octadecyl silica gel column chromatography using a water–acetonitrile solvent system to give fraction F with water–acetonitrile (0:1) as the eluent. Fraction F was subjected to preparative TLC with hexane–isopropanol (199:1) to give magnumiol (**3**; 0.6 mg). Data for **3** are as follows: colorless oil; ^1^H NMR and ^13^C NMR spectroscopic data are shown in Table 3; and HR-FABMS *m/z* 381.2380 [M^+^Na]^+^ (381.2403 calculated for C_23_H_34_O_3_Na).

### 4.6. Cell Proliferation Assay

K562 cells were maintained at 37 °C (5% CO_2_ in air) in tissue culture dishes filled with RPMI1640 medium (Sigma-Aldrich, St. Louis, MO, USA) supplemented with 10% (*v*/*v*) fetal bovine serum (FBS), 25 μg/mL penicillin, and 50 μg/mL streptomycin (RPMI–FBS). HeLa and 3T3-L1 cells were maintained at 37 °C (5% CO_2_ in air) in tissue culture dishes filled with Dulbecco’s modified Eagle’s medium (DMEM; Sigma-Aldrich) supplemented with 10% (*v*/*v*) FBS, 25 μg/mL penicillin, and 50 μg/mL streptomycin (DMEM–FBS). For the cell proliferation assay, K562 cells (2 × 10^4^ cells/well), HeLa, and 3T3-1 cells (5 × 10^3^ cells/well) were incubated for three days in 12-well plates, with each well containing 1 mL of RPMI–FBS (for K562 cells) or DMEM–FBS (for HeLa and 3T3-L1 cells) and the additives in duplicate (the additives were 0.2% (*v*/*v*) dimethyl sulfoxide and 20–40 μM of **1** or **2**). The relative cell number was assessed using Alamar as a cell number indicator (Fujifilm Wako Pure Chemical Corporation, Osaka, Japan), and the half-maximal inhibitory concentrations (IC_50_) of **1** and **2** were determined as described previously [28,29].

### 4.7. Measurement of the Minimum Inhibitory Concentration (MIC)

The Gram-positive bacteria methicillin-susceptible *Staphylococcus aureus* (MSSA; strain ATCC29213), methicillin-resistant *S. aureus* (MRSA; ATCC43300), and the Gram-negative bacterium *Escherichia coli* (ATCC25922) were suspended in Mueller–Hinton broth (5 × 10^5^ CFU/mL; 0.1 mL/well) and incubated for 24 h at 37 °C in 96-well plates (Corning, New York, NY, USA) in the presence of a vehicle, serially diluted test compounds at various concentrations, or known antibiotics (oxacillin, vancomycin, and ampicillin) [30]. MIC was defined as the lowest concentration of the additives that inhibited visible bacterial growth.

### 4.8. Calculation of the EDC Spectrum of Intermedipyrone (***2***)

Conformational searches and density functional theory (DFT) calculations were conducted using Avogadro software [31] and the Gaussian 16 program [32], respectively. Intermedipyrone (**2**) was submitted to conformational searches using the molecular mechanics MMFF94s. The initial stable conformers with Boltzmann distributions over 1% were further optimized via DFT calculations at the B3LYP/6-31G(d) level. The stable conformers with Boltzmann distributions over 1% were subjected to time-dependent DFT calculations at the B3LYP/6-31G+(d,p) level in the presence of acetonitrile with a polarizable continuum model. The resultant rotatory strengths of the lowest 30 excited states for **2** were converted into Gaussian-type curves with half-bands (0.2 eV) using GaussView 6 software [33].

## Figures and Tables

**Figure 1 molecules-29-02143-f001:**
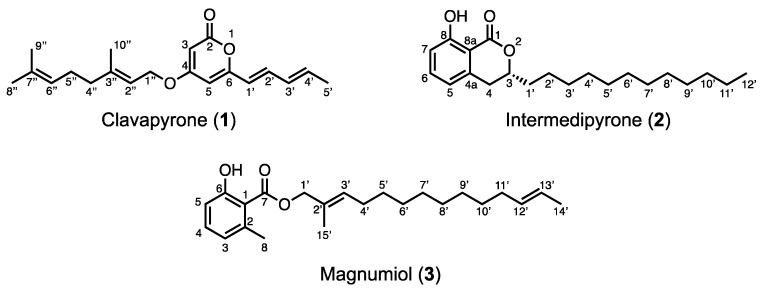
Structures of clavapyrone (**1**), intermedipyrone (**2**), and magnumiol (**3**).

**Figure 2 molecules-29-02143-f002:**
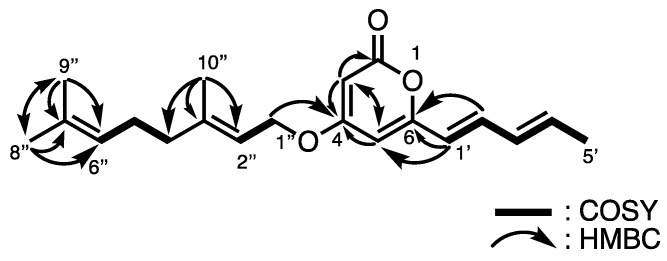
Structure of **1** and representative ^1^H–^1^H COSY and HMBC correlations.

**Figure 3 molecules-29-02143-f003:**
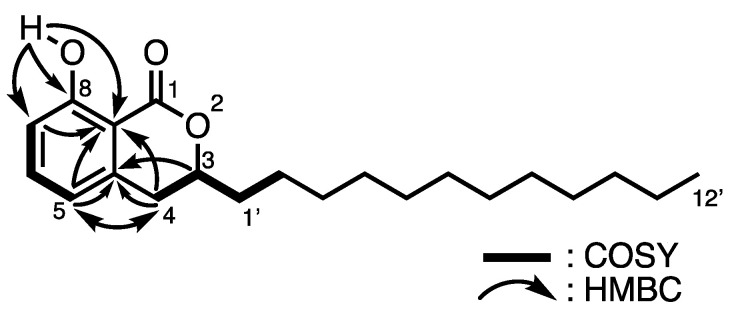
Structure of **2** and representative ^1^H–^1^H COSY and HMBC correlations.

**Figure 4 molecules-29-02143-f004:**
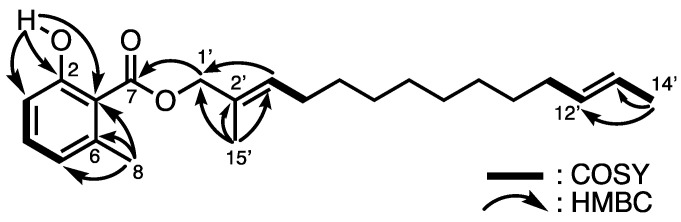
Structure of **3** and representative ^1^H–^1^H COSY and HMBC correlations.

**Figure 5 molecules-29-02143-f005:**
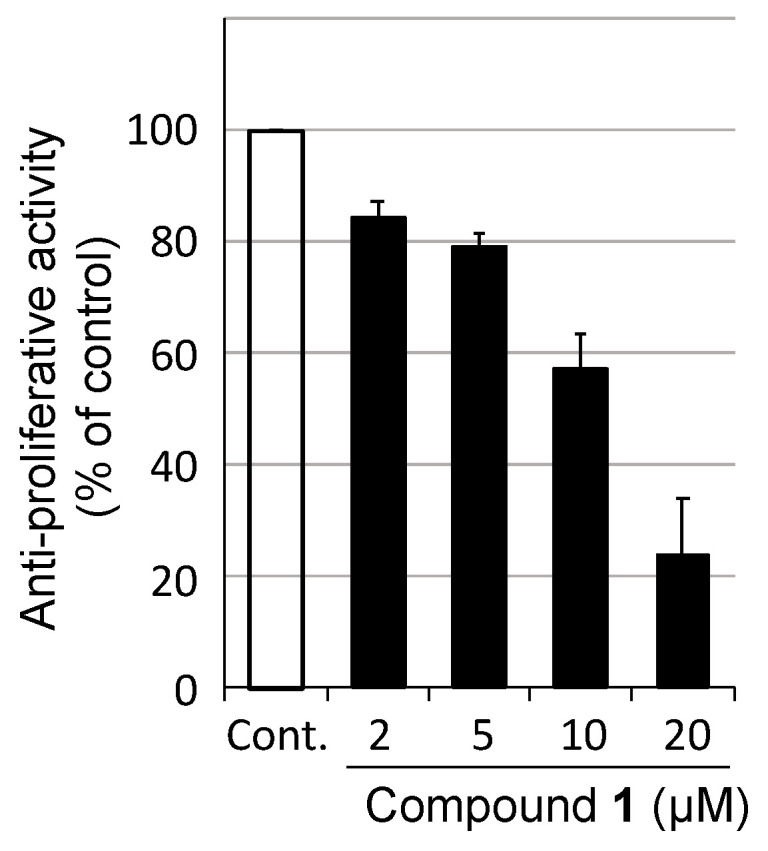
Antiproliferative activity of **1** against K562 cells.

**Table 1 molecules-29-02143-t001:** NMR spectral data of pyrone (**1**) ^1^.

Position	^13^C (ppm)	^1^H (ppm)	
1	-	-	
2	164.3	-	
3	89.1	5.42	(d, *J* = 2.1 Hz, 1H)
4	170.4	-	
5	100.8	5.79	(d, *J* = 2.1 Hz, 1H)
6	159.1	-	
1′	119.8	5.92	(d, *J* = 15.3 Hz, 1H)
2′	136.5	7.08	(dd, *J* = 15.3, 10.9 Hz, 1H)
3′	130.5	6.15	(ddt, *J* = 15.3, 10.9, 0.6 Hz, 1H)
4′	136.9	6.03	(dd, *J* = 15.0, 7.1 Hz, 1H)
5′	18.7	1.83	(dt, *J* = 7.1, 0.6 Hz, 3H)
1″	65.8	4.50	(d, *J* = 6.7 Hz, 2H)
2″	117.2	5.38	(ddt, *J* = 6.7, 2.4, 1.1 Hz, 1H)
3″	143.6	-	
4″	39.5	2.05–2.09	(m, 2H)
5″	26.3	2.09–2.14	(m, 2H)
6″	123.6	5.05	(tt, *J* = 6.7, 1.0 Hz, 1H)
7″	132.2	-	
8″	25.7	1.64	(d, *J* = 1.3 Hz, 3H)
9″	17.8	1.58	(s, 3H)
10″	16.8	1.70	(s, 3H)

^1^ 600 MHz for ^1^H and 150 MHz ^13^C in CDCl_3_.

**Table 2 molecules-29-02143-t002:** NMR spectral data of intermedipyrone (**2**) ^1^.

Position	^13^C (ppm)	^1^H (ppm)	
1	170.0	-	
2	-	-	
3	79.8	4.54–4.60	(m, 1H)
4	32.9	2.91–2.94	(m, 2H)
4a	139.5	-	
5	117.9	6.89	(d, *J* = 8.4 Hz, 1H)
6	136.1	7.40	(dd, *J* = 8.4, 7.5 Hz, 1H)
7	116.2	6.69	(dd, *J* = 7.5, 0.4 Hz, 1H)
8	162.2		
8a	108.5		
8-OH	-	11.0	(s, 1H)
1′	34.8	1.69–1.76	(m, 1H)
		1.84–1.92	(m, 1H)
2′	24.8	1.42–1.50	(m, 1H)
		1.49–1.60	(m, 1H)
3′–9′	29.3	1.23–1.38	(m, 14H)
	29.3		
	29.5		
	29.5		
	29.6		
	29.6		
	29.6		
10′	31.9	1.17–1.37	(m, 2H)
11′	22.7	1.17–1.37	(m, 2H)
12′	14.1	0.85	(t, *J* = 7.0 Hz, 3H)

^1^ 600 MHz for ^1^H and 150 MHz ^13^C in CDCl_3_.

**Table 3 molecules-29-02143-t003:** NMR spectral data of magnumiol (**3**).^1.^

Position	^13^C (ppm)	^1^H (ppm)	
1	112.5	-	
2	162.9	-	
3	115.6	6.84	(dd, *J* = 8.1, 0.6 Hz, 1H)
4	134.1	7.27	(t, *J* = 8.1 Hz, 1H)
5	122.9	6.71	(dd, *J* = 8.1, 0.6 Hz, 1H)
6	141.3	-	
7	171.7	-	
8	24.2	2.55	(s, 3H)
2-OH	-	11.4	(s, 1H)
1′	71.9	4.75	(s, 2H)
2′	129.0	-	
3′	131.8	5.58	(t, *J* = 7.3 Hz, 1H)
4′	27.8	2.04–2.09	(m, 2H)
5′–10′	29.2	1.20–1.40	(m, 12H)
	29.2		
	29.5		
	29.5		
	29.6		
	29.6		
11′	32.6	1.93–1.97	(m, 2H)
12′	131.7	5.40–5.42	(m, 1H)
13′	124.6	5.40–5.42	(m, 1H)
14′	17.9	1.64	(m, 3H)
15′	14.4	1.74	(s, 3H)

^1^ 600 MHz for ^1^H and 150 MHz ^13^C in CDCl_3_.

## Data Availability

Dataset available on request from the authors.

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
