# Peer review of "Isolation and Structure Determination of New Pyrones from Dictyostelium spp. Cellular Slime Molds Coincubated with Pseudomonas spp."

_molecules, 2024, doi:10.3390/molecules29092143_

Round 1

Reviewer 1 Report

Comments and Suggestions for Authors

This manuscript describes isolation, structure elucidation and biological activities of three new compounds from the fruiting bodies of cellular smile molds cultured with Pseudomonas spp..In the compound’s identification part, structures of all three compounds were determined based on the combination of HREIMS and NMR analyses, and explanation is logical, easy to follow. This work also demonstrates that cellular slime molds coculture with bacteria approach is very useful for discover new molecules. Overall this is an interesting paper which is suitable to publish on the Molecules .

As my concern: The authors state that these compounds are predicted to be biosynthesized by type-I iterative polyketide synthases in the abstract, however the experimental basis for this assessment is not investigated and therefore should probably be removed from the abstract or at least make it clear in the manuscript. Besides, in my opinion, compounds 1 and 3 are more like meroterpenoid than polyketide which may be biosynthesized by the combination of an isoprenoid with polyketide. Then the statement is also not accurate.

Author Response

Thank you for your helpful comments. In this paper, there is no experimental data to follow biosynthetic routes for isolated compounds. However, pyrone moiety and 6-methylsalicylic acid are known to be biosynthesized by type-I PKS. So, we removed the sentence about biosynthetic routes and rewrote the sentence as follows.

In the abstract, “These compounds are not obtained from usual cultural conditions with Klebsiella aerogenes.”

In the Discussion section, “6-Alkylpyrone moiety in clavapyrone and 6-methylsalicylic acid in intermedipyrone are predicted to be biosynthesized by type-I iterative polyketide synthases (iPKS), which are typical of fungi [25, 26].”

Reviewer 2 Report

Comments and Suggestions for Authors

Summary

This paper resarces the impact of different feeding bacteria on the secondary metabolite production in cellular slime molds. The authors isolated three specific compounds (clavapyrone, intermedipyrone, and magnumiol) from various Dictyostelium species. Their findings suggest that the diversity of feeding bacteria can significantly expand the range of secondary metabolites produced, highlighting the potential of cellular slime molds for the discovery of new pharmaceuticals.

Main Contributions and Strengths

  • Novelty: The exploration of different feeding bacteria for secondary metabolite production is relatively new in cellular slime mold research.
  • Potential Applications: The results suggest broadened pharmaceutical applications for cellular slime molds.

General Concept Comments

  • Testability: The hypothesis, that different bacterial strains influence secondary metabolite production, is testable. A more robust study design with various bacteria would further strengthen conclusions.
  • Missing Controls: The paper lacks a comparative study of secondary metabolites produced when feeding the slime molds their typical diet of Klebsiella aerogenes. This control is needed to fully demonstrate the impact of alternative bacteria.
  • Scope: While the findings are interesting, the study analyzes a limited number of Dictyostelium species and compounds. Expanding the sample size would provide a stronger basis for the conclusions.

Please find the comments in a PDF file attached to this message. 

Best regards, 

Author Response

---Missing Controls: The paper lacks a comparative study of secondary metabolites produced when feeding the slime molds their typical diet of Klebsiella aerogenes. This control is needed to fully demonstrate the impact of alternative bacteria.

---Scope: While the findings are interesting, the study analyzes a limited number of Dictyostelium species and compounds. Expanding the sample size would provide a stronger basis for the conclusions.

Thank you for your helpful comments. We attempted to identify changes in the extracts cultured with Klebsiella aerogenes and Pseudomonas spp. respectively. However, isolated compounds could not be observed by LC ESI-MS and we could not compare differences. We speculate that this was because the amount of the compound produced by the cellular slime mold was very small, the concentration in the extract was below the detection limit, and the compounds were highly hydrophobic and could not be ionized by ESI.

In our preliminary investigations using Klebsiella aerogenes, these compounds have not been isolated, but we cannot prove that we couldn’t isolate them because these compounds are undetectable on HPLC.

As this reviewer says, we should show objectively the differences in production by changing feeding bacteria and increase the sample size. To reflect this, the text has been rewritten and added as follows.

“Furthermore, since bacteria other than Pseudomonas spp. can be used to culture cellular slime molds [12, 13], and the number of species of cellular slime molds and feeding bacteria will be increased to demonstrate the usefulness of this organism for drug discovery. In addition, studies are currently underway using techniques such as Molecular Networking [27] to objectively demonstrate differences in extracts due to changes in feeding bacteria, which will accelerate the exploration of novel compounds.”

---Please find the comments in a PDF file attached to this message.

Thank you for your pointing out. The sections pointed out in the PDF have been corrected as follows.

In line 62. We added a reference.

In line 73, 115, and 149. Methanol extracts were filtered and evaporated, and then partitioned. So, the sentences were rewritten as follows. In addition, the duplicated parts with the method section (In line 233, 247, and 261) have been simplified.

“After extracting twice with methanol at room temperature, filtration, and evaporation, the resulting extract (15.5 g) was partitioned between ethyl acetate and water.”

---Reviewer Comments: Which drug was considered as control? Also , the comparison of the effect of this compounds against a known treatment drug will be even more interesting. Is there any particular reason why this cellular line was just used? Why not any others?

As reviewer 2 pointed out, we should have added some known drugs as a positive control. However, the purpose of our research this time was not to compare the strength of antitumor (antiproliferative) activity between existing drugs and the newly found compounds, but to examine whether the new compounds have antitumor activity within a pharmacological concentration range. In fact, we found that compound 1 has antitumor activity, so please accept our results without a positive control.

  There are several reasons for using K562 human leukemia cells. This cell line is widely used to study the antitumor activity of new compounds, and we have often utilized it. The fact that these cells are human-derived and that they are easy to handle as they are floating cells is also attractive.

---Reviewer Comments: Control used? (line 293)

In line 293. We added the sentence about controls as follows.

“….. in the presence of a vehicle, serially diluted test compounds at various concentrations, or known antibiotics (oxacillin, vancomycin, and ampicillin) [30].”

---About reference 32 (33 in the revised manuscript)

This reference is for GaussView 6 software and is in the recommended citation format.

Reviewer 3 Report

Comments and Suggestions for Authors

In this work, three cellular slime molds Dictyostelium spp. were co-cultured with three bacteria Pseudomonas. Interestingly, three new pyrones 13 were yielded, respectively. Their structures were elucidated by extensive analysis of spectral data. Furthermore, the absolute configuration of compound 2 was determined by CD helicity rule and calculations, respectively. Cytotoxic and antibacterial bioassays were carried out and the results showed that compound 1 displayed antiproliferative activity against human leukemia K562 cells. These findings were important, especially disclosed the effects of different feeding bacteria on the production of secondary metabolites from cellular slime molds.

However, some concerns as following:

1. In this work, cellular slime molds Dictyostelium spp. were co-cultured with bacteria of the genus Pseudomonas, but the related information was missing in the Abstract.

2. Please provide the HRMS spectra of these new compounds in Supplementary Materials.

3. It is better to provide the 1H NMR data when analyze the presence of some groups, which could not be identified unambiguously by the 13C NMR data. For instance, four methyl groups [δC 18.7 (C-5’) and δH 1.83 (dt, J = 7.1, 0.6 Hz, 3H, H-5’), δC 25.7 (C-8’’) and δH 1.64 (d, J = 1.3 Hz, 3H, H-8’’), δC 17.8 (C-9’’) and δH 1.58 (s, 3H, H-9’’), and δC 16.8 (C-10’’) and  δH 1.64 (s, 3H, H-10’’)].

4. What was the positive control for the bioassays? And since the cellular slime molds were co-cultured with bacteria, was it necessary to evaluate the antibacterial activity for the isolated compounds?

5. It is better to discuss the different structural features of secondary metabolites from cellular slime molds co-incubated with different bacterial reported by other groups.

Others:

1. ‘NMR spectra data’ →‘NMR spectral data’

2. ‘1H NMR spectra’ →‘1H NMR spectrum’

3. Please provide the retention times (Rt) for compounds 1 and 2, which were purified by HPLC.

Author Response

  1. In this work, cellular slime molds Dictyostelium spp. were co-cultured with bacteria of the genus Pseudomonas, but the related information was missing in the Abstract.

Thank you for pointing out. We rewrote the sentence in Abstract as follows (line 26).

“These compounds are not obtained from usual cultural conditions with Klebsiella aerogenes, but obtained from coincubated conditions with Pseudomonas spp.”

  1. Please provide the HRMS spectra of these new compounds in Supplementary Materials.

Thank you for pointing out. We added HRMS spectra in the Supplementary Material.

  1. It is better to provide the 1H NMR data when analyze the presence of some groups, which could not be identified unambiguously by the 13C NMR data. For instance, four methyl groups [δC 18.7 (C-5’) and δH 1.83 (dt, J = 7.1, 0.6 Hz, 3H, H-5’), δC 25.7 (C-8’’) and δH 1.64 (d, J = 1.3 Hz, 3H, H-8’’), δC 17.8 (C-9’’) and δH 1.58 (s, 3H, H-9’’), and δC 16.8 (C-10’’) and δH 1.64 (s, 3H, H-10’’)].

All carbons and protons are assigned each other by HMQC spectra and they are summarized in the tables. We considered that it was not necessary to provide the 1H NMR chemical shifts in the text unless they were essential for structure determination.

  1. What was the positive control for the bioassays? And since the cellular slime molds were co-cultured with bacteria, was it necessary to evaluate the antibacterial activity for the isolated compounds?

We should have added some known drugs as a positive control. However, the purpose of our research this time was not to compare the strength of antitumor (antiproliferative) activity between existing drugs and the newly found compounds, but to examine whether the new compounds have antitumor activity within a pharmacological concentration range. In fact, we found that compound 1 has antitumor activity, so please accept our results without a positive control.

 We added the sentence about controls for antibacterial assays as follows.

“….. in the presence of a vehicle, serially diluted test compounds at various concentrations, or known antibiotics (oxacillin, vancomycin, and ampicillin) [30].”

Since the amount of compounds produced by cellular slime molds is very small, the antibacterial assay was tested with high concentrations. In addition, cellular slime molds are thought to have a growth phase by feeding bacteria and a production phase under starvation after the bacteria have been consumed. Even if the compounds produced had antimicrobial activity, it is not yet known whether they would have an effect during incubation. Therefore, we believe that it was worthwhile to measure the antimicrobial activities of the isolated compounds.

  1. It is better to discuss the different structural features of secondary metabolites from cellular slime molds co-incubated with different bacteria reported by other groups.

To our best knowledge, there are no groups focused on the difference of secondary metabolites of cellular slime molds by feeding bacteria. Therefore, it is difficult to compare with other groups.

However, our group previously isolated pyrones from cellular slime molds (ref.6, 8). In addition, we continuously focused on this organism, and we proposed that cellular slime molds may take up compounds produced by coincubated bacteria (ref.3). The relationship between these and isolated compounds is currently under investigation.

Others:

  1. ‘NMR spectra data’ →‘NMR spectral data’
  2. ‘1H NMR spectra’ →‘1H NMR spectrum’
  3. Please provide the retention times (Rt) for compounds 1 and 2, which were purified by HPLC

Thank you for pointing out. We corrected for 1 and 2 (Table 1, 2, 3, line 158). For 3, in this study, we used a recycle HPLC system, so that we can’t provide the retention times.